# Levels of Moral Distress among Health Care Professionals Working in Hospital and Community Settings: A Cross Sectional Study

**DOI:** 10.3390/healthcare9121673

**Published:** 2021-12-03

**Authors:** Noemi Giannetta, Rebecca Sergi, Giulia Villa, Federico Pennestrì, Roberta Sala, Roberto Mordacci, Duilio Fiorenzo Manara

**Affiliations:** 1Faculty of Philosophy, Vita-Salute San Raffaele University, 20132 Milan, Italy; noemigiannetta93@gmail.com (N.G.); sergi.rebecca@unisr.it (R.S.); pennestri.federico@hsr.it (F.P.); sala.roberta@unisr.it (R.S.); mordacci.roberto@unisr.it (R.M.); 2Center for Nursing Research and Innovation, Vita-Salute San Raffaele University, 20132 Milan, Italy; manara.duilio@hsr.it

**Keywords:** moral distress, healthcare professionals, community setting, hospital setting

## Abstract

Moral distress is a concern for all healthcare professionals working in all care settings. Based on our knowledge, no studies explore the differences in levels of moral distress in hospital and community settings. This study aims to examine the level of moral distress among healthcare professional working in community or hospital settings and compare it by demographic and workplace characteristics. This is a cross-sectional study. All the professionals working in the hospitals or community settings involved received personal e-mail invitations to participate in the study. The Moral Distress Thermometer was used to measure moral distress among healthcare professionals. Before data collection, ethical approval was obtained from each setting where the participants were enrolled. The sample of this study is made up of 397 healthcare professionals: 53.65% of the sample works in hospital setting while 46.35% of the sample works in community setting. Moral distress was present in all professional groups. Findings have shown that nurses experienced level of moral distress higher than other healthcare professionals (mean: 4.91). There was a significant differences between moral distress among different professional categories (H(6) = 14.407; *p* < 0.05). The ETA Coefficient test showed significant variation between healthcare professionals working in community and in hospital settings. Specifically, healthcare professionals who work in hospital experienced a higher level of moral distress than those who work in community settings (means 4.92 vs. means 3.80). The results of this study confirm that it is imperative to develop educational programs to reduce moral distress even in those settings where the level perceived is low, in order to mitigate the moral residue and the crescendo effect.

## 1. Introduction

The healthcare system is characterized by an important heterogeneity of services and users that make it complex. The evolution of demographic dynamics, the increase in chronic diseases, and the rationalization of resources are just some of the factors that led to profound changes in the health care system, followed by repercussions on the operators who assist the people daily. Indeed, more and more literature in the field of medical ethics is devoted to the moral distress [1,2]. Moral distress was first theorized in the health care and nursing field in 1984, when Jameton [3] defined it as a human condition that arise when “one knows the right thing to do, but institutional constraints make it nearly impossible to pursue the right course of action”.

Early studies on this topic were confined to the intensive care setting [4,5], specifically related to end-of-life treatment [6,7]. However, empirical research on this topic has seen exponential growth in recent years, intensifying the study of moral distress in any other clinical care setting (long-term care facilities, pediatric or neonatal settings and emergency or mental health departments) [8,9,10].

Studies have shown that all healthcare professionals are exposed to moral distress across daily assistance. Dodek and Johnson-Coyle [11,12,13] have shown that nurses in clinical field had higher moral distress scores than physicians or other healthcare professionals. Maybe, higher moral distress scores were associated with lower degrees of decisional attitude and social support. However, most of the studies reported no statistically significant differences in moral distress scores among healthcare professionals.

Research on moral distress triggers and consequences is growing up to the point that some authors have considered defining moral distress as an “umbrella term” [14,15]. To curb this confusion, Morley [16] performed a narrative synthesis of the moral distress literature, identifying moral judgment, psychological and physical effects, moral dilemmas, moral uncertainty, external and internal constraints, and threats to moral integrity as necessary conditions for the occurrence of moral distress. Furthermore, according to literature, the main cause of moral distress could be related to patient-level factors, unit/team level factors, and system-level factors in hospital settings [10,17,18,19]. Indeed, studies have shown that healthcare professionals are exposed to frequent conditions causing moral distress. For example, performing painful procedures or futile care, following physicians’ prescription or family wishes which are considered inconsistent with the patient’s good, perceiving unsafe or incompetent staffing [20,21].

Austin [22] has shown that most of nurses tend to leave their position after they face with moral distress. A positive correlation between turnover intention and moral distress was found by Hashemi et al. [23]. Other factors can lead to moral distress, including heavy workload, poor communication with team members or patient and its relatives, limited resources (e.g., staff shortage) [24,25].

This situation can affect the quality of care perceived by the patient, increase institutional costs and length of hospital stay. Furthermore, moral distress is a trigger of headache, digestive disorders, anger, feeling of guilt, job dissatisfaction, burnout, and depression [26,27,28]. With regards to healthcare professionals’ job satisfaction and satisfaction of workplace, Soodabeh [29] described a significant correlation between job satisfaction and moral distress. When nurses experience a higher moral distress score, they have also a low job satisfaction. Several studies confirm these findings in hospital setting but little is known about job satisfaction levels among healthcare workers working in community setting, and its correlation between moral distress.

Despite the evidence available on triggers and consequences of moral distress among nurses or healthcare workers in hospitals and community settings, most of studies aimed to describe the intensity and frequency of moral distress only in a specific setting. Based on our knowledge, there were no study aimed to compare levels of moral distress in different settings. However, even if there were some differences in everyday care in hospital or community setting, a scoping review has recently shown that the main triggers of moral distress among healthcare professionals in community setting are the same identified in hospital setting: a poor organization of working process, conflicting interpersonal relationship, lack of trust from patient or family members [8]. More detailed research is needed in order to describe the differences in levels of moral distress among healthcare professionals working in community or hospital settings. In addition, it is not known which characteristics of community or hospital workplace and which characteristics of healthcare professionals are independently associated with moral distress. Therefore, the purpose of this study was to determine the relationship between moral distress in Italian community and hospital setting and their demographic and professional characteristic and workplace characteristics.

## 2. Materials and Methods

### 2.1. Aim

This study aimed to examine the level of moral distress among healthcare professional working in community or hospital settings and compare moral distress levels by demographic and workplace characteristics.

### 2.2. Design and Setting

This study used a cross-sectional survey design. The study reporting was supported by the STROBE checklist (Appendix A). A convenience sample was recruited. All healthcare professionals who practice in hospitals or community settings received personal e-mail invitations to participate in the study. After 30 days from the first email, a second reminder was sent. The e-mail invitation to potential participants included information about the aims of the study, a letter of informed consent, and the link to the online survey. Healthcare professionals had to meet the following inclusion criteria: Working in hospital or community settings providing care for the elderly with a job experience of more than 6 months; willingness to participate in the study; knowledge and understanding of the Italian language. Exclusion criteria were students or non-professional caregivers and healthcare professionals with maximum six month of job experience. Data collection took place from February to May 2021. Data were collected in two community setting and one hospital of Northern Italy.

### 2.3. Measurements

To collect data the authors used a questionnaire made up of three sections. A copy of the questionnaire is available upon request.

The first section aimed to collect the socio-demographic characteristics of healthcare professionals (gender, nationality, mother tongue, age, schooling level, social status) and professional characteristics (working setting (hospital or community setting), whether if they work during COVID-19 pandemic, full-time or part-time work, professional category, working experience, time and kilometers taken to travel from home to work).

The second section aimed to collect information about the workplace where each healthcare professional worked. Based on a scale from 0 to 10, healthcare professionals have specified the importance given to communication in their profession; how much are they satisfied with their position; how much are they satisfied with their salary; how much they feel the needs of the elderly are met; how much are they satisfied with workplace characteristic (silence, pleasant spaces, privacy, comfort for patients’ movement). To create this section, a focus group is made up. Expert on community and hospital care was included in order to evaluate the workplace characteristic that are different among settings and that could be predictive of moral distress. After the development of each item, the pilot study was made up and the reliability was assessed. This section has a Cronbach’s α of 0.883 for a mixture of Italian respondents, assessed in this study.

The third section aimed to measure moral distress using the Moral Distress Thermometer (MDT). This is a screening tool to measure moral distress based on visual analogue scales and verbal numeric rating scales [30]. This instrument with 0–10 rating scales was used to describe how much moral distress has been experiences. Moreover, it defines moral distress as “not perceived”, “light”, “uncomfortable”, “distressing”, “intense”, “worst possible”. Based on Wocial’s results [30], this instrument had a good convergent validity and concurrent validity. Its Cronbach’s α was adequate (0.90).

### 2.4. Ethical Considerations

Before data collection, ethical approval was obtained from each setting where the participants were enrolled. All of them participated voluntarily, were informed about the study aims and procedures, and were informed about their right to participate and withdraw at any time. All participants signed informed consent. The researcher also provided each participant a unique identification code to guarantee data protection and anonymity.

### 2.5. Data Analysis

Statistical analysis has been run using SPSS^®^ statistical software, version 26 (IBM Corp: Armonk, NY, USA). Socio-demographic and professional characteristics were analyzed using the descriptive statistical analysis (means, standard deviation (SD), frequency, and percentages). In order to test the normality of the data and to decide which type of test to adopt, the Skewness and Kurtosis and the Shapiro–Wilk test were calculated. The Shapiro–Wilk test for normality shows that *p* value is below 0.05, so null hypothesis must be rejected and distributions cannot be considered normal. In this study, the distribution of items was not normal. Based on this observation, non-parametric analysis was adopted. A Mann–Whitney U Test was used to determine the statistical difference between the means of hospital and community setting on workplace characteristics. A Kruskal–Wallis test was used to test the difference between professional categories and moral distress.

The ETA test was used to determine the association strength between workplace (community or hospital setting) and moral distress levels. The significance level was set at *p* < 0.05. Finally, Cronbach’s alpha coefficient is calculated. A general accepted rule is that a Cronbach alpha >0.70 represents an acceptable level of reliability of the tests, while 0.80 and above represents a strong reliability. Cronbach’s alpha for healthcare professionals’ conditions suggested a strong reliability of the measures, up to 0.88.

## 3. Results

### 3.1. Characteristics of the Sample

477 healthcare professionals working in hospital and community settings were invited to take part to the survey. Of these, 10 did not give consent to participate and 77 did not complete all the questionnaire. Therefore, the entire sample of this study was made up of 397 healthcare professionals: 53.65% of the sample was working in a hospital setting, while 46.35% of the sample was working in community settings. The majority of the sample was female (*n* = 290; 73.0%) and Italian (*n* = 318; 80.1%). The mean age of the sample was 42.27 ± 10.78 years with a range of 22–67. With regard to the professional area, the majority of the sample (*n* = 91; 35.4) were nurses; 18.9% were certified nursing assistants (in Italian, “operatore socio-sanitario” or “ausiliario socio-assistenziale”); 18.3% were physicians; the rest 7.8% physical therapists and 2.0% educators. Therefore, it was possible to categorize all of these professionals in four macro-categories: 59.14% in healthcare, 34.01% in tutelary assistance, 2.79% in entertainment and socialization activities, and the remaining 4.06% in “other”, various area. The mean length of the sample working experience is 16.50 ± 10.43 years, with a range of 1–42. Table 1 shows the demographic and professional characteristics of the sample based on workplace.

### 3.2. Characteristics of Work Activity and Workplace

Communication between patient and healthcare professionals was considered very important by most of the sample included in this study (mean: 9.16; SD ± 1.29). On a scale from 1 to 10, healthcare professionals have expressed their opinions about workplace comfort in terms of silence (mean: 6.00; SD ± 2.37), pleasant spaces (mean: 6.28; SD ± 2.28), space available (mean: 6.23; SD ± 2.37), privacy (mean: 6.14; SD ± 2.53), and comfort for patients’ movements (mean: 6.14; SD ± 2.49). Table 2 shows the degree of agreement on these items based on healthcare working in community setting, hospital setting, and overall.

Findings show that healthcare professionals working in hospital setting had lower averages than those working in community settings (see Table 2). The Mann–Whitney U Test has shown the statistical difference between the two groups’ means (see Table 2). Indeed, difference is statistically significant for all the variables pointed out in Table 2. Thus, the null hypothesis for Mann–Whitney U can be rejected and it can be stated that nurses working in hospital setting had lower averages with regard to silence (U = 12,607.500; z = −6.046; *p* < 0.001), pleasant spaces (U = 10,241.00; z= −8.217; *p* < 0.001), space available (U = 9145.000; z = −9.127; *p* < 0.001), privacy (U = 9775.000; z = −8.564; *p* < 0.001), and comfort for patients’ movements (U = 6533.500; z = −11.445; *p* = *p* < 0.001) than colleagues working in the community setting.

Table 3 shown further details about the statistical test conduct.

### 3.3. Level of Moral Distress

Moral distress was assessed using the Moral Distress Thermometer. Of the 397 healthcare professionals who responded to the Moral Distress Thermometer (mean = 4.40; SD ± 2.715), 179 (45.2%) reported their level of moral distress as uncomfortable or higher. Moreover, 7.32% of the respondents stated that they have not perceived any kind of moral distress for the two weeks prior the questionnaire, while 6.06% of the sample reported they have felt it the worst possible way. Table 4 shows the levels of moral distress among healthcare professionals working in community or hospital settings.

Table 5 shows the levels of moral distress based on professional role. Nurses have the highest level of moral distress (mean: 4.91). As shown in Table 6 and according to Kruskal–Wallis test, there is a statistically significant association between professional categories and moral distress (H(6) = 14.407; *p* < 0.05).

### 3.4. Moral Distress and Work Characteristics

The next analysis examined the relationship among workplace characteristics and moral distress. To provide significance of such, it has been proceeded with a Spearman correlation for non-parametric tests. These findings are summarized in Table 7. In general, the perceived level of moral distress was higher in those professionals reporting lower degrees of satisfaction with salary (*r* = −0.356; *p* < 0.001) and job position (*r* = −0.283; *p* < 0.001); lower attention to the needs of the elderly (*r* = −0.31; *p* < 0.001); higher desire to change job (*r* = 0.269; *p* < 0.001) or workplace (*r* = 0.358; *p* < 0.001); lower satisfaction with respect to places (silence (*r* = −0.324; *p* < 0.001), pleasant spaces (*r* = −0.374; *p* < 0.001), privacy (*r* = −0.374; *p* < 0.001), space available (*r* = −0.407; *p* < 0.001), and comfort for movements (*r* = −0.418; *p* < 0.001)). 

It is also interesting to note that there is no significant correlation between moral distress and the first three variables in the table: distance from home to the workplace in terms of communication within the workplace (*r* = −0.009; *p* > 0.05).

### 3.5. Moral Distress in Hospital and Community Settings

An ETA coefficient test was also run in order to determine the strength of association between Moral Distress and settings (community and hospital). Moral Distress was chosen as the dependent and findings are statistically significant for settings (community and hospital) and Moral Distress.

## 4. Discussion

This study examined moral distress in hospital and community settings, which is an area that had not been previously explored in the literature. Even if technical care provided in different setting are the same, the purpose of care and the associated ethical issues are generally different among hospital and community settings [8]. However, Moral Distress is a concern for all the healthcare professionals working in all care settings. Based on this, this study aimed to examine the level of moral distress among the healthcare professionals working in community or hospital settings and compare them by demographic and workplace characteristics.

With regards to job category, moral distress was present in all professional groups. Previous studies on moral distress have shown that it was prevalent among all healthcare providers. A recent study conducted in Tehran has shown that nurses working in hospital have higher levels of Moral Distress [31], while Almutairi et al. [32] have reported a moderate level of Moral Distress.

We found that nurses experienced a higher level of Moral Distress in comparison to other healthcare professionals. These findings agree with those of Whitehead et al. [33], according to which nurses had significantly higher levels than physicians and auxiliary healthcare professional. This may be due to the nature of nursing science and its “professional responsibility” of providing direct patient care. In addition, closer contact and proximity to the patient make nurses more exposed to morally distressing events and consequently increase the level of moral distress level at baseline.

Healthcare professionals included in this study confirmed that moral distress exists in community and hospital care setting. Findings show a significant difference between healthcare professionals working in community and hospitals settings (*p* < 0.05). Specifically, healthcare professionals working in hospital settings experienced a higher level of moral distress than those working in community settings (4.92 vs. 3.80). Based on our knowledge, this is the first study aimed to compare the levels of moral distress among different setting. Indeed, Wocial et al. [34] analyzed the level of moral distress among physicians and found that they were less likely to experience moral distress when caring for patients living in a nursing home. However, Wocial et al. [34] did not compare these findings with the hospital setting. Indeed, in this study most of the sample working in community setting reported a low to moderate range of moral distress perceived, while most of the sample working in hospital settings reported a distressing to worst possible range. Even if the level is low, moral distress may accumulate over time and may give rise to moral distress residue crescendo. This concept was first studied by Epstein & Hamric [35] who explored the effect of moral distress over time and its consequences on healthcare professionals and patients: even if the event that caused moral distress among a certain healthcare professional is solved, a moral residue can remain, raising up the exposition level baseline. Given that, our findings suggest that interventions are needed to help prevent crescendo effects in healthcare professionals working in community and hospital settings.

Moreover, moral distress seems to arise from several sources that could be distinguished into personal and workplace characteristics. According to our findings, the perceived level of moral distress was higher in those professionals reporting lower degrees of satisfaction with salary and job position; lower attention to the needs of the elderly; higher desire to change job or workplace; lower satisfaction with respect to places (silence, pleasant spaces, privacy, space available and comfort for movements).

These findings agree with recent studies and reviews with the purpose of identifying the determinants of moral distress in care settings [4,36]. These results provide an indication of the impact of specific workplace characteristic on moral distress scores in healthcare professionals. In our study, means scores related to workplace satisfaction are higher in community setting than hospital setting. Healthcare professionals working in hospital setting show a greater propensity to leave a job position or change activities than professionals in community setting. These variables are associated with moral distress levels (*p* < 0.05). Indeed, the study conducted by Karagozoglu et al. [18] shows that 24.0% of the nurses considering leaving job position due to moral distress, as well as the study conducted by Fernandez-Parsons et al. [37] show that most of nurses hope to be transferred to another unit due to moral distress. These and our findings confirmed a relationship between moral distress and organizational issues [10,17,18,19].

Moreover, means of satisfaction with regard to silence, pleasant spaces, privacy, space available, and comfort for movements are higher among nurses working in community setting than nurses working in hospital setting. A recent study had shown that physical environmental features are important for the development of a “sense of home”, especially in long-term care settings [38].

Based on these considerations, it would be recommended to adopt interventions on the organizational culture and institutional framework of hospital and community settings. However, any change in organizational culture must be followed by introducing dedicated educational programs into the nursing and healthcare professionals curricula.

According to Austin, “although the term, moral distress, is not part of our ordinary language, using it can help us speak to the moral domain of our practice” [39]. It is necessary to sensitize all healthcare professionals to the phenomenon of moral distress so that everyone can recognize it early in daily practice and take action to prevent it or solve it. In addition, nurse managers should also be aware of it. Indeed, healthcare professionals report higher levels of moral distress when their managers are closed and “insensitive” to the phenomenon. Moreover, literature shows that communication with patient, its relatives, and colleagues is key to managing or downsizing misunderstandings at all levels, decreasing levels of moral distress [40].

In order to sensitize all healthcare professionals, in a recent conference, it was proposed to change the term “moral distress” with “moral resilience”, giving to this term a positive meaning and encouraging the identification of preventive more than “therapeutic” arrangements to address it [41]. It should be noted that to date there is no cornerstone tool to help policy decision makers identify healthcare professionals at risk of moral distress at an early stage, and propose specific solutions [42]. Further studies are recommended to develop an instrument that “diagnose moral distress”.

Although the participants in this study were invited from email, one limitation of this study is that they all worked in hospital or community settings in the Northern Italy, therefore they may not be generalizable in terms of Moral Distress levels perceived in other community or hospital settings. At the same time, all the participants involved in this study work in a Region of Northern Italy, the Lombardy, where aging is particularly pronounced, and investments through the enhancement of community care and the coordination of hospital and community care professionals are being experimented to the benefit of elderly patients [43]. The administration of the questionnaire via online did not allow tracking of response rates. However, we collect the total number of healthcare professionals working in the setting included in this study. Furthermore, we decided to send a second reminder, after 30 days from the first email, to increase the participation to the survey.

Moreover, the present study did not consider any psychological and personological variables such as religion that might represent confounding variables. Therefore, further studies that explore these variables are recommended.

## 5. Conclusions

Moral distress is a prevalent condition among all healthcare professionals. Although previous studies confirm the existence of moral distress in all care settings, based on our knowledge, this is the first study aimed to explore difference in the level of moral distress among different healthcare settings. Findings have shown different levels of moral distress between different settings: the healthcare professionals working in a hospital setting reported higher means of moral distress than those who work in community settings.

However, moral distress was present in all professional groups. We found that nurses experienced a higher level of Moral Distress in comparison to other healthcare professionals. According to our findings, the perceived level of moral distress was higher in those professionals reporting lower degrees of satisfaction with salary and job position; lower attention to the needs of the elderly; higher desire to change job or workplace; lower satisfaction with respect to places (silence, pleasant spaces, privacy, space available, and comfort for movements).

These findings confirm that it would be recommended to adopt interventions on the organizational culture and institutional framework of hospital and community settings.

Pathways that help health professionals to feel more satisfied with their job position as well as the creation of working groups in which communication between the members of the team is fostered are highly recommended strategies based on the results of our study.

However, any change in organizational culture must be followed by introducing dedicated educational programs. It is imperative to develop educational programs to reduce moral distress even in those settings where healthcare professionals experience low levels of moral distress, to mitigate the moral residue and the crescendo effect. Further qualitative or quantitative studies are recommended to specifically identify the sources of moral distress and investigate them in different settings.

## Figures and Tables

**Table 1 healthcare-09-01673-t001:** Demographics and professional characteristics of the sample.

Variable	Community Setting—*n* (%)	Hospital Setting—*n* (%)
Gender (Male vs. Female)	45 (24.5)/139 (75.5)	62 (29.1)/151 (70.9)
Age	Mean: 43.29; SD: 9.98	Mean: 41.41; SD: 11.367
Education		
High School Diploma	113 (61.4)	23 (10.8)
Bachelor’s Degree	52 (28.3)	120 (56.3)
Master’s degree	19 (10.3)	63 (29.6)
PhD	-	7 (3.3)
Nationality		
Italian	120 (65.2)	198 (93.0)
Other	64 (34.8)	15 (7.0)
Marital status		
Single	60 (32.6)	61 (28.6)
Engaged/Married	96 (52.2)	136 (63.8)
Divorced	27 (14.7)	15 (7.0)
Widower	1 (0.5)	1 (0.5)
Sons		
Yes	117 (63.6)	107 (50.2)
No	67 (36.4)	106 (49.8)
Professional Area		
Social worker in public health service	66 (59.4)	9 (6.2)
Nurses	13 (7.1)	78 (53.4)
Physician	7 (6.3)	40 (27.4)
Physiotherapy	12 (10.8)	8 (5.5)
Educator	6 (5.4)	2 (1.4)
Other (psychologist; speech therapist; orthoptist)	7 (6.3)	9 (6.2)
Work experience as professionals (Mean in years; SD)	Mean: 16.15; SD: 10.76	Mean: 16.79; SD: 10.16
Work experience in care setting for elderly (Mean in years; SD)	Mean: 13.82; SD: 8.75	Mean:14.36; SD: 9.69
Mean of distance from home to workplace (kilometers)	Mean: 14.06; SD: 13.58	Mean: 17.24; SD: 24.28
Mean of distance from home to workplace (minutes)	Mean: 29.22; SD: 21.12	Mean: 27.74; SD: 21.45
In the last couple of weeks, how many times have I thought about changing professional activity?	Mean: 0.81; SD: 2.19)	Mean: 2.33; SD: 5.62
In the last couple of weeks, how many times have I thought about changing workplace?	Mean: 1.46; SD: 2.91)	Mean: 3.32; SD: 6.66
Mean of the sample’s degree of agreement on importance of communication	Mean: 9.10; SD: 1.29	Mean: 9.22; SD: 1.30
Mean of satisfaction on work position (range 0–10)	Mean: 7.23; SD: 1.93	Mean: 7.01; SD: 1.82
Mean of salary satisfaction (range 0–10)	Mean: 5.75; SD: 2.27	Mean: 4.95; SD: 2.11

**Table 2 healthcare-09-01673-t002:** Mean of the sample’s degree of agreement on comfort of workplace.

Variable	Community Setting	Hospital Setting	Total Sample	Mann–Whitney U
Silence	6.81 (1.94)	5.31 (2.48)	6.00 (2.37)	12,607.500; z = −6.046; *p* = 0.000
Pleasant spaces	7.28 (1.99)	5.43 (2.17)	6.28 (2.28)	10,241.000; z = −8.217; *p* = 0.000
Space available	7.39 (1.92)	5.24 (2.27)	6.23 (2.37)	9145.000; z = −9.127; *p* = 0.000
Privacy	7.33 (1.98)	5.12 (2.52)	6.14 (2.53)	9775.000; z = −8.564; *p* = 0.000
Comfort for patients’ movement	7.65 (1.77)	4.85 (2.27)	6.14 (2.49)	6533.500; z = −11.445; *p* = 0.000

**Table 3 healthcare-09-01673-t003:** Mean ranks for Mann–Whitney U test, following Table 2 results.

Variable	Context	N	Mean Rank	Sum of Ranks
Silence	Community setting	182	235.23	42,811.50
	Hospital setting	213	166.19	35,398.50
Pleasant spaces	Community setting	182	249.04	45,574.00
	Hospital setting	213	155.08	33,032.00
Space available	Community setting	182	254.25	46,274.00
	Hospital setting	213	149.93	31,936.00
Privacy	Community setting	182	250.79	45,644.00
	Hospital setting	213	152.89	32,566.00
Comfort for patients’ movement	Community setting	182	268.60	48,885.50
	Hospital setting	213	137.67	29,324.50

**Table 4 healthcare-09-01673-t004:** Level of moral distress among healthcare professionals.

Variable	Community Setting *n* (%)	Hospital Setting *n* (%)
Moral distress level	Mean: 3.80—SD: 2.70	Mean: 4.92—SD: 2.62
Moral distress level per moral distress thermometer categories		
Greater than distressing to worst possible (>6–10)	38 (20.7)	64 (30.0)
Greater than uncomfortable to distressing (>4–6)	28 (15.3)	49 (23.0)
Greater than mild to uncomfortable (>2–4)	47 (25.7)	58 (27.2)
Less than or equal to mild (0–2)	70 (38.3)	42 (19.7)

**Table 5 healthcare-09-01673-t005:** Levels of moral distress based on professional role.

Professional Role	Not Perceived *n* (%)	Light *n* (%)	Uncomfortable *n* (%)	Distressing *n* (%)	Intense *n* (%)	Worst Way Possible *n* (%)	Mean
Certified nursing assistants—OSS	6 (2.33)	11 (4.28)	7 (2.72)	7 (2.72)	6 (2.33)	6 (2.33)	4.35
Certified nursing assistants—ASA	3 (1.16)	4 (1.55)	11 (4.28)	4 (1.55)	7 (2.72)	3 (1.16)	4.56
Nurses	6 (2.33)	15 (5.83)	14 (5.44)	27 (10.50)	26 (10.11)	3 (1.16)	4.91
Doctors	1 (0.38)	9 (3.50)	14 (5.44)	10 (3.89)	10 (3.89)	3 (1.16)	4.74
Physiotherapists	1 (0.38)	4 (1.55)	7 (2.72)	3 (1.16)	3 (1.16)	2 (0.77)	4.35
Educators	-	-	5 (1.94)	2 (0.77)	1 (0.38)	-	4.50
Others	2 (0.77)	8 (3.11)	5 (1.94)	1 (0.38)	-	-	2.31

**Table 6 healthcare-09-01673-t006:** Association between professional categories and moral distress (*p* < 0.05).

	Value
Valid N	257
Test statistic	14.407 a
Degrees of freedom	6
Asymptotic Sig. (2-sided)	0.025

a. The test statistic is adjusted for ties.

**Table 7 healthcare-09-01673-t007:** Spearman correlations of workplace characteristics with moral distress levels.

Variable	Correlation Coefficient	*p* Value	N
Communication	−0.009	0.864	396
Satisfaction on position	−0.283 **	0.000	396
Salary satisfaction	−0.356 **	0.000	395
Elderlies’ needs	−0.312 **	0.000	395
Change activity	0.269 **	0.000	392
Change workplace	0.358 **	0.000	392
Silent spaces	−0.324 **	0.000	395
Pleasant spaces	−0.374 **	0.000	396
Space available	−0.407 **	0.000	395
Privacy	−0.374 **	0.000	395
Comfort for movements	−0.418 **	0.000	395

** Correlation coefficient that are significant (*p* < 0.01).

## Data Availability

All data are available upon request.

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
