# Peer review of "Levels of Moral Distress among Health Care Professionals Working in Hospital and Community Settings: A Cross Sectional Study"

_healthcare, 2021, doi:10.3390/healthcare9121673_

Round 1

Reviewer 1 Report

Dear authors, I congratulate you for writing this study. I point out some suggestions / considerations for improving the article.
Introduction: very well written and logical triggering of ideas. QUESTION: the written paragraph (line 70 to 76) is not comprehensive. "Austin [22] has shown that most nurses tend to leave their position after they face 70
moral distress. The positive correlation between turnover intention and moral distress was 71
found by Hashemi et al. [23]. Other factors can lead to moral distress, including heavy 72
workload, low nurse-to-patient ratios, poor communication with team members or pa- 73
tient and its relatives [24; 25]."- WHAT WOULD IT BE: low nurse-patient ratio? I suggest revising that paragraph.

The method is very well written and supported in STROBE. I suggest that you further detail the inclusion and exclusion criteria. QUESTION: which statistical test used to test the normality of the data? Specify and justify.

The results are adequate and reflect the robust methodology outlined above. Regarding the discussion, I miss comparisons with other studies, and, above all, criticism of the few compared studies.
QUESTION: given the limitations presented, what are the potentials to minimize the limitations described?
About the conclusion, be objective. Highlight the main findings.
once again congratulations on writing.

Author Response

Dear reviewer

thanks for your precious suggestion and notes.

Attached our comments

Regards

Reviewer 2 Report

This is an interesting and useful study, and after suitable revision deserves to be published. I have the following technical comments, which imply a need for change or improvement.

(1) The "decimal point" is frequently presented in the European style as a comma (although not always). For the sake of clarity, and to meet the needs of an international readership, I suggest that a period:  "." be consistently used to indicate a decimal point.  Occasionally references in brackets are separated by a semi-colon, rather than a comma - MDPI's house style should prevail here.

(2)  The STROBE questionnaire/data is said to be downloadable (but not for this reviewer). It would strengthen the paper if this could be added to the published article as an Appendix.

(3)  In section 2.3 (lines 123-4) the intriguing question is presented: " ... whether if they have sons or not and eventually how many".  Now, I am sure that in Italy as elsewhere, daughters and sons are equally valuable, so why ask only about sons (and future sons)?  Since this question did not (apparently) yield any significant variation, I suggest that it be omitted, since it is an idiosyncratic and rather embarrassing question!

(4) The variable "satisfaction with retribution" yields significant results. One trembles at contemplating the exact nature of such retribution, since it implies a high level of moral outrage, and punishment -  and perhaps divine sanction for wrong-doing in healthcare settings.  Do the authors actually mean "remuneration"  (salary levels). We need clarification!

(5)  The use of Spearman's Rho is sound, when some of the variables do not have normal distribution (e.g. as categorical variables). Use of Eta Squared is good also, but does require a normally distributed dependent variable in the ANOVA model.  More explanation on assumptions about statistical design, appropriate statistical variables, and their interpretation would be useful.

Author Response

Dear reviewer

thanks for your precious suggestion and notes.

Attached our comments

Regards

This manuscript is a resubmission of an earlier submission. The following is a list of the peer review reports and author responses from that submission.

Round 1

Reviewer 1 Report

The issues discussed by the authors of the article are cognitively valuable, necessary because of their application value. The article submitted for review is prepared correctly. The literature that the authors refer to is up-to-date. However, the research tool used - the questionnaire in its first section, requires the provision of data proving the methodological correctness, at least in terms of reliability. The note relates to questions requiring evaluation "the importance given to communication in their profession; how much are they satisfied with their position; how much are they satisfied with their retribution; how much they feel the needs of the elderly are met"(88 -91 lines).

Author Response

Thanks for your comments and suggestions. Attached are the notes to the reviewer.

Thanks

Reviewer 2 Report

Thank you for the opportunity to read your Manuscript titled "Levels of moral distress among health care professionals working in hospital and community setting: a cross sectional study". I find the article interesting, but I have a few rather some comments. 

The Introduction is too concise and not focused on hypothesis; moreover, the study hypothesis and study motivation should be presented more in details. 

I suggest to verify the method of reporting the p value (eg. p<0.01)

Have any confounding variables been considered (eg, religion, psychological or personological variables)?

In the analysis section, have you considered conducting ANOVA?

The value of the article could also be increased by extending the Discussion section with a discussion of the practical Implications of the results and development of the literature implemented in the introduction.

Author Response

(The authors gave the same response as above.)

Reviewer 3 Report

Thank you for the opportunity to review this manuscript. This is a highly important topic that seems to have been made even more important due to the pandemic. I am from the United States and nursing/hospital workers are leaving the profession in droves due to many reasons including moral, physical health, and mental well-being being compromised due to the significant asks of them all right now with the hospitals so full. That being said, I do think this article needs a lot of work and would not recommend it to be published. Do keep in mind, while I do feel qualified to review the current manuscript in terms of its theoretical rigor and analytic strategies/conclusions, I have not submitted work to this journal previously and am not overly familiar with the structure of articles at this journal. I am simply voicing my concerns with the manuscript from a general manuscript submission perspective.

Major concerns:

The introduction does a pretty good job of setting up the problem with real world issues but is nearly devoid of theoretical rationale for these issues. I would like to see more done to theoretically discuss the issues regarding morality for healthcare workers and how this affects the main issues the article is addressing/studying. The results also include extra variables that are not well set up in the introduction.

My largest issue is the framing of the results. Substantial time is spent discussing characteristics of the sample rather than spending the valuable journal space discussing the main findings with regard to the research question/hypotheses. Yes, this information is valuable, but might I suggest supplementary material?

Similar to the above point, I am losing the main point through the results. It took me a few reads to really grasp the findings with regard to the hypotheses. Even now, though, confusion remains. For example, the authors state that there was no significant differences between the professional categories and moral distress but give a significant p value. I’m assuming the differences weren’t significant except for the “other” category? I’m just not certain what is happening here.

The discussion section largely just reiterates the findings. I’d like to see more done to discuss where we go from here, the theoretical implications, etc.

Minor concerns

Much of the manuscript was written in present tense while manuscripts are typically framed in past tense.

There are a lot of typos, grammatical errors, etc. throughout the manuscript. For example, even in the title the word “hospital” has two h’s. Care should be taken to correct these issues.

Author Response

(The authors gave the same response as above.)

Round 2

Reviewer 2 Report

Thanks for the editing, in my opinion it is adequate.

Author Response

Thanks for your supervision and your comment.